# Risk Prediction Models for Oral Cancer: A Systematic Review

**DOI:** 10.3390/cancers16030617

**Published:** 2024-01-31

**Authors:** Aufia Espressivo, Z. Sienna Pan, Juliet A. Usher-Smith, Hannah Harrison

**Affiliations:** Department of Public Health and Primary Care, University of Cambridge, Cambridge CB2 0SR, UK; zp270@cam.ac.uk (Z.S.P.); jau20@medschl.cam.ac.uk (J.A.U.-S.); hh504@medschl.cam.ac.uk (H.H.)

**Keywords:** oral cancer, risk prediction, risk models, screening, primary care

## Abstract

**Simple Summary:**

Oral cancer is among the twenty most common cancers worldwide. Finding and treating this cancer early improves survival rates. Screening the whole population to check for oral cancer is unlikely to be an efficient use of resources; however, screening only individuals at higher risk has been shown to reduce oral cancer deaths and be cost-effective for healthcare services. Mathematical models have previously been developed to identify these high-risk groups; however, it is not known whether any of these would be suitable for use in clinical practice. In this study, we identified and compared previously published models. We found several that had potential, but only two had been tested outside the original study population. We suggest that future research should focus on (a) testing how well the models identify those at high risk within potential screening populations and (b) assessing how the models might be included within the healthcare systems.

**Abstract:**

In the last 30 years, there has been an increasing incidence of oral cancer worldwide. Earlier detection of oral cancer has been shown to improve survival rates. However, given the relatively low prevalence of this disease, population-wide screening is likely to be inefficient. Risk prediction models could be used to target screening to those at highest risk or to select individuals for preventative interventions. This review (a) systematically identified published models that predict the development of oral cancer and are suitable for use in the general population and (b) described and compared the identified models, focusing on their development, including risk factors, performance and applicability to risk-stratified screening. A search was carried out in November 2022 in the Medline, Embase and Cochrane Library databases to identify primary research papers that report the development or validation of models predicting the risk of developing oral cancer (cancers of the oral cavity or oropharynx). The PROBAST tool was used to evaluate the risk of bias in the identified studies and the applicability of the models they describe. The search identified 11,222 articles, of which 14 studies (describing 23 models), satisfied the eligibility criteria of this review. The most commonly included risk factors were age (*n* = 20), alcohol consumption (*n* = 18) and smoking (*n* = 17). Six of the included models incorporated genetic information and three used biomarkers as predictors. Including information on human papillomavirus status was shown to improve model performance; however, this was only included in a small number of models. Most of the identified models (*n* = 13) showed good or excellent discrimination (AUROC > 0.7). Only fourteen models had been validated and only two of these validations were carried out in populations distinct from the model development population (external validation). Conclusions: Several risk prediction models have been identified that could be used to identify individuals at the highest risk of oral cancer within the context of screening programmes. However, external validation of these models in the target population is required, and, subsequently, an assessment of the feasibility of implementation with a risk-stratified screening programme for oral cancer.

## 1. Introduction

Oral cavity and lip cancer together accounted for the 16th highest rate of both cancer incidence and mortality globally in 2020, with 377,713 new cases and 177,757 deaths recorded that year [1]. Additionally, the prevalence of oral cancer varies between countries [2,3], with around two-thirds of cases occurring in lower- to middle-income countries [4]. Between 1990 and 2017, the global age-standardised rate of incidence has increased from 4.41 to 4.84 per 100,000 person-years, while the age-standardised rate of mortality (2.4 per 100,000 person-years) and disability-adjusted life years (64.0 per 100,000 person-years) have remained unchanged [5]. The increasing incidence and stable mortality rates suggest an improvement in treatment strategies, potentially partially due to advances in surgical techniques [5,6]. Early detection with appropriate treatments has been shown to improve overall survival rates (90% five-year survival rate in those with oral potentially malignant disorders [OPMD]) who had a follow-up, compared with 56% in patients without known OPMD) as well as lowering the rate of recurrence [7,8,9,10,11,12]. Currently, most patients (over 60%) tend to be diagnosed at later stages of the disease when it has spread [13], highlighting the difficulty of diagnosing this cancer at early stages when it is often asymptomatic, especially in populations where routine dental check-ups are not part of standard healthcare [4,9,14].

The low disease prevalence, estimated to be between 0.12 and 4.12 per 1000 in lower- and middle-income countries, means that population-wide screening is unlikely to be an efficient method to improve early detection rates of oral cancer [15,16,17]. As yet, no country has yet implemented a systematic national population-based screening programme for oral cancer [15]. A recent analysis investigating if oral cancer met the criteria for a national population screening programme in the United Kingdom [18] identified a number of gaps in the current evidence, including a lack of suitable methods to identify people with high-risk lesions. It has been suggested that screening targeted towards high-risk groups may effectively reduce mortality [14,15], and there is evidence that this approach may be more cost-effective [19]. Taiwan has been implementing a screening programme targeting high-risk populations since 2004 [20,21], where individuals with smoking or betel quid chewing habits are invited to screening [21].

A prediction model could be used to assess the risk of developing oral cancer for individuals in a population and identify those at the highest risk to be targeted for screening. The ideal model for use within a risk-stratified screening programme would use risk factors easily obtainable through routine clinical practice. Further, the model would have to be shown to perform well in the target population [22,23]. A recent rapid review [24] identified a number of models that were developed to predict the risk of oral cancer; however, this was not comprehensive and it is unclear if any are suitable for use within a screening programme. This review aims to (a) systematically identify published models that predict the development of oral cancer that are suitable for use in the general population and (b) describe and compare the identified models, considering their development, including risk factors, performance and their applicability within the context of risk-stratified screening.

## 2. Materials and Methods

We performed a systematic review following an a priori established study protocol (PROSPERO ID: CRD42022316516).

### 2.1. Search Strategy

We performed a literature search in Medline, Embase and the Cochrane Library, with no language restrictions, from the beginning of the database up to November 2022, to identify studies on the development and/or validation of risk prediction models for oral cancer. A combination of the following subject headings was used: ‘oral cancer’, ‘risk factor/risk assessment/risk’ and ‘prediction/model/score’ (for full search strategy, see Appendix A). The reference lists of all included articles were also manually searched to identify other relevant studies.

### 2.2. Selection of Included Studies

We included studies that fulfilled the following criteria: (a) is a primary research paper in a peer-reviewed journal; (b) provides a measure of risk for the development of primary oral cancer that incorporates two or more risk factors acting at an individual level; (c) is applicable to adults in the general population; (d) reports a quantitative measure of model performance. We excluded studies (a) including only specific groups of the population, for example, long-term tobacco users or patients with premalignant oral lesions, and (b) reporting models that predict disease progression.

There is no consensus on the definition of the umbrella term “oral cancer” [25], either clinically or within research. In this review, we define oral cancer as all cancers affecting the oral cavity or oropharynx. This includes cancers of the lip, oral cavity (upper- and lower-lip mucosa, tongue, gingiva, floor of the mouth, hard palate, buccal mucosa, vestibule and retromolar area) and oropharynx (ICD-10: C00-06, C09, and C10), which all have similar biology, aetiology and several common risk factors, such as smoking and alcohol consumption [26,27]. In the case where the study did not specify the type of oral cancer, we categorised the outcome as oral cavity cancer (OCC). We did not include studies with an outcome of nonspecific forms of head and neck cancer or cancer of the upper aerodigestive tract; these may include (but are not limited to) cancers of the oral cavity.

#### Screening Process

One reviewer (A.E.) performed the search and deduplicated the identified articles. All retrieved articles were imported into EndNote 20 (Clarivate, London, UK), which was used for citation management and deduplication [28]. Three reviewers (A.E., Z.S.P., H.H.) independently screened 10% of all articles by title and abstract (including pilot screening) in Rayyan (Rayyan Systems, Cambridge, MA, USA) [29]; all disagreements were resolved through group discussion. The remaining 90% of articles were then screened by two reviewers (A.E., Z.S.P.).

The full text was examined if a decision to exclude could not be made based on the title and abstract alone. All full texts were assessed independently by two reviewers (A.E., Z.S.P.). Disagreements between the reviewers were resolved through discussion between the two reviewers or consultation with a third reviewer (H.H., J.A.U.-S.) when a consensus could not be reached.

### 2.3. Data Extraction

Data extraction of all included studies was carried out independently by two reviewers (A.E., Z.S.P.) using a standardised data extraction form (Appendix A). Information about the model development (population and statistical methods), the published model (risk factors included) and the performance of the model in both development and validation (including discrimination and calibration) was extracted from all included studies. The studies were classified using the Transparent Reporting of a Multivariable Prediction Model for Individual Prognosis or Diagnosis (TRIPOD) guidelines [30,31] and we assessed both their risk of bias and applicability to a risk-stratified screening programme using the Prediction Model Risk of Bias Assessment Tools (PROBAST) across four domains (population, risk factors, outcomes, and analysis) [32,33]. PROBAST tool is a comprehensive tool, published in 2019 following several rounds of expert consultation, which provides a robust assessment of the risk of bias for each individual risk model and enables the identification of areas where the overall research quality is low.

In cases where multiple models are reported for the same population and outcome (for example, different combinations of risk factors are used within a stepwise selection process) only the model with the best performance was extracted. However, all models were extracted separately for studies that report more than one distinct model: for different subgroups of the population (for example, separate models for men and women), for different outcomes (e.g., oral cancer, oropharyngeal cancer [OPC]) or using different risk factors (for example, comparing performance of a model with and without a biomarker).

## 3. Results

### 3.1. Study Selection

After duplicates were removed, the literature search identified 11,222 articles, of which 10,676 were excluded by title and abstract screening. Of the remaining 546 articles, the full-text screening excluded 533 articles. The most common reasons for exclusion were not reporting quantitative performance measures (*n* = 375) or not predicting the risk of developing oral cancer for individuals (*n* = 73) (Appendix A).

One paper was identified through citation searching and included at the full-text level. The full study selection process is shown in Figure 1. Overall, 14 studies, corresponding to 23 models, are included in the data synthesis [34,35,36,37,38,39,40,41,42,43,44,45,46,47].

### 3.2. Model Development and Validation

A summary of the 23 models is presented in detail in Table 1 (phenotypic-only models) and Table 2 (models including genetic risk factors).

The most common outcome is OCC (*n* = 18) [34,35,37,38,40,41,44,45,46,47], which includes all identified models with included genetic information (*n* = 6) [35,44,45,46,47]. The remaining models were developed for OPC (*n* = 4) [41,43] or the composite outcome OCC or OPC (*n* = 2) [42]. Two studies [37,41] developed separate models for men and women.

The majority of risk models were developed or validated in populations from China (*n* = 9) and the United States (*n* = 6) (Appendix A). All models were developed in case-control studies, except for one [39] that was developed in a cluster randomised controlled trial (RCT)-based cohort study [48]. Most models were developed in populations recruited from hospital settings (*n* = 12) [34,35,37,38,40,42,44,45,47]. In three studies (corresponding to seven models) the cases were hospital-based and controls were recruited from either the general population (*n* = 2) [43] or a combination of the hospital- and community-based populations (*n* = 5) [36,41]. The studies by Cheung et al. [39] and Fritsche et al. [46] developed models using recruited cases and controls from the general population.

Only three models were developed to estimate the absolute risk of developing oral cancer [39,41,43]. Most models (*n* = 22) were developed using logistic regression, with one developed using survival analysis (Cox proportional hazards) [39].

Fourteen of the models have been internally validated: seven using resampling within the development population [37,39,42,46], four using a random split-sample [41] and one using a nonrandom split-sample [44]. Two models were also validated in an external population [43].

### 3.3. Risk of Bias in Studies

All models were assessed as having a high overall risk of bias (Appendix A) with the most common issues in the population and analysis domains. In the population domain, 21 out of 23 development studies and 12 out of 14 validation studies were assessed to have a high risk of bias. The most common reason for this was the use of case-control study designs for model development (*n* = 22). Models developed using study designs at a lower risk of bias, for example, the cohort study by Cheung et al. [39], might have had a lower risk of bias; however, in this case, they did not report clear or objective eligibility criteria for the control participants. In the analysis domain, 16 out of 23 model development studies and 7 out of 14 model validation studies were assessed to have a high risk of bias. This was most commonly due to insufficient reporting of the performance measures of the models (*n* = 19) and/or the use of univariable analysis (*n* = 17) to select the included risk factors.

In this review, a low score for concerns about applicability indicates that the model is suitable for predicting the risk of an individual developing oral cancer in the general population. However, all of the models had a high or unclear overall score for concerns regarding applicability. Most concerns were identified in the population domain, where 13 models were given a high rating (due to the use of hospital-based populations) and seven models were given an unclear rating (due to the use of mixed hospital-based and general population).

### 3.4. Risk Factors

Across all the included studies, 55 non-genetic risk factors were considered for inclusion in a model, of which 48 were included in at least one model and 31 were included in two or more models (Figure 2 and Table 3).

Most of the models included at least one demographic or lifestyle risk factor (*n* = 21) and most included two or more (*n* = 18). The most common demographic and lifestyle risk factors were age (*n* = 20), alcohol consumption (*n* = 18), smoking (*n* = 17) and education level (*n* = 14). Sex was included in nine of the 17 models developed using mixed-sex cohorts. Nine models also included a risk factor for a family history of any cancer [36,37,40,41,42].

Seven models included clinical risk factors, most commonly markers of oral health or reporting of oral habits (*n* = 6) [36,37,39]. These include tooth loss, recurrent oral ulceration, regular dental visits, denture wearing, oral rinsing habits and OCC screening status (Table 3). One model included human papillomavirus (HPV) status as a variable [43]. Three models included a blood-based biomarker (serum levels of Arsenic, Cerium and Selenium) along with demographic and lifestyle risk factors [35,38,40]; none of these biomarkers are currently used within routine clinical practice.

Six models used genetic risk factors [40,44,45,46,47], all in combination with demographic and lifestyle risk factors, and one [35] additional included a biomarker (Selenium level). The genetic models included between two and 1,119,238 single nucleotide polymorphisms (SNPs) (Appendix A). Four genetic models [35,44,45,47] used small numbers of SNPs (2–7 SNPs) previously shown to be associated with biological mechanisms for oral cancer development. For example, Chung et al. (2017) [44] developed genetic risk scores (GRS) for OCC using four SNPs previously shown to be associated with oral cancer in betel quid users. The two models developed by Fritsche et al. [46] derived polygenic risk scores (PRS) using SNPs that had been shown to be associated with oral cancer in genome-wide association studies (GWAS). All six of these genetic models used logistic regression to combine genetic and phenotypic risk factors.

### 3.5. Model Performance

The most widely used measure of discrimination is the area under the receiver-operating curve (AUROC). The receiver-operating curve is used to plot sensitivity against 1-specificity for a range of cut-off points. An AUROC of 1 indicates that, in the cohort used to test the model, the model always assigns a higher score to any individual who goes on to develop kidney cancer than it assigns to any individual who does not develop kidney cancer. An AUROC of 0.5 indicates the model does not perform better than random chance or flipping a coin. A measure of discrimination, such as the AUROC values or C-statistic, was reported for 18 models (Figure 3, Table 4). Reported discrimination ranged from an AUROC of <0.55 to 0.95, with heterogeneity across the models and between those within each risk factor group (Figure 3). This suggests that the difference in performance was driven more by the variation in the study designs and study populations (Appendix A) than by the risk factors themselves. The two models with the highest reported discrimination were the models developed by Tota et al. [43] and Chung et al. [45], and they, respectively, included clinical and genetic risk factors (in addition to demographics and lifestyle risk factors) and had AUROCs of 0.95 in internal validation [43] and 0.91 in the development population, respectively [45]. Two studies reported models from the same population with incrementally increasing risk factors. In two models by Tota et al. [43], including information on HPV status improved the performance both in internal (AUROC 0.86 to 0.95) and external validation (AUROC 0.81 to 0.87). Conversely, in the case-control study by He et al. [40] with the development population of 324 oral cancer patients and 650 disease-free controls, adding a biomarker (Cerium level) to a model containing demographic and lifestyle risk factors did not result in an improvement in discrimination (AUROCs 0.78 and 0.77, respectively).

An odds ratio (OR) between different risk groups was reported as a measure of performance for five of the six models, which included genetic risk factors (Table 4). We note the performance of the model developed by Chung et al. (2017) [44] (a GRS calculated using four SNPs adjusted for age and betel quid chewing status) with an OR of 3.11 (95% CI: 1.21–10.67) for individuals with four alleles. The PRS developed by Fritsche et al. [44] showed limited ability to identify people at the highest risk of developing oral cancer (ORs for the top 1% of the population 1.63 [95% CI: 0.81–3.26] and 1.69 [95% CI: 0.61–4.71] for the two models—each developed in slightly different populations (Appendix A). The highest OR per unit score was observed in the model by Chung et al. (2019) [44] for participants with a categorical GRS of 2 (participants having two alleles associated with oral cancer, OR: 6.12 [95% CI: 1.66–22.49]) compared to those with the lowest GRS of 0 as reference.

Measures of model accuracy were reported for three models [38,44,45], all of which were developed in hospital-based populations (Appendix A). The reported sensitivity and specificity ranged from 69.9% to 92.8% and from 60.3% to 91.4%, respectively. Models with the highest sensitivity and specificity were Rao 2016b [42] and Chen 2022 [38]. Only one study by Rao et al. [42] reported positive or negative predictive values (PPV or NPV), with PPVs of 77.3% and 51.4%, respectively, for the internally validated multivariable model (2016a) and risk score (2016b).

A measure of calibration was reported for 15 of the 23 risk models. Nine reported quantified measures, four reported AIC values (these are not directly comparable to models developed in other populations), and six reported calibration graphically. These graphical measures were reported in studies by Chen et al. (2018) [37], corresponding to two development models, and by Lee et al. [41], corresponding to four separate-sex models for OCC and OPC, which showed good calibration in an internal validation except for one model (men, OPC) that overestimated the risk for higher deciles. Only one study, Tota et al. [43], reported model calibration in an external validation, with O/E ratios of 1.08 (compared to 1.01 for the same model in an internal validation).

## 4. Discussion

### 4.1. Key Findings

To our knowledge, this is the first systematic review of risk prediction models for oral cancer. We have identified multiple models, which have been developed to predict the risk of individuals in the general population developing oral cancer.

The identified models use a wide range of risk factors, including clinical, genetic and blood-based biomarkers in addition to demographics and lifestyle risk factors. Although the reported discrimination of the models was wide-ranging (AUROCs 0.53–0.95), we identified nine models with AUROC > 0.7 in a validation, including the two models with AUROC > 0.8 in external validations.

The performance of the models is consistent with that found for models predicting the risk of developing other cancers in earlier systematic reviews for kidney cancer [49], breast cancer [50] and colorectal cancer [51]. Similarly to kidney cancer [52], there has been very limited development of PRS for oral cancer, with the only two models identified in this review (both developed by Fritsche et al. [44]) including a PRS. However, the studies by Chung (2017) et al. [44], which incorporate small numbers of SNPs examining the interaction between genetics and smoking behaviour, have shown promising results. The externally validated models, developed by Tota et al. [43], show very promising results, especially when combining HPV status with demographic and lifestyle risk factors.

However, the heterogeneity of the study populations, model development methods and risk factors considered in the studies included in the review, and the general lack of external validations, make direct comparisons between the models or risk factors challenging.

### 4.2. Model Generalisability

Only two of the identified models (both developed by Tota et al. [43]) have been externally validated [43]. Additionally, we did not identify any studies that modelled the expected impact of the models in a clinical scenario (for example, the proportion of cases a model would be able to identify within a risk-stratified screening programme). We note that, unlike in similar reviews in other cancers, most of the models identified in this review were developed in populations drawn from lower- and middle- income countries. This may reflect the relatively low prevalence of oral cancer in higher-income countries. Although the highest age-standardised incidence of oral cancer is seen in South Asian countries [5,53], some of the models (eight out of 23) we identified were developed and validated in populations from the USA and the UK, where incidence is lower. For example, the models by Tota et al. [43], which were developed and externally validated in North American populations, would require validations in target screening populations before implementation in the South Asian countries. Similarly, the PRS developed by Fritsche et al. [46], restricted their development cohorts to individuals with European ancestry, therefore, their performance (reported discrimination is poor) is likely to be worse in other populations. Those models that were developed in low-income countries with a high incidence of oral cancer, (for example, the models developed by He et al. [40] and Rao et al. [42] in China and India, respectively), use small, hospital-based populations, and generalisability to wider populations has not been tested.

### 4.3. Availability of Risk Factors

A key consideration for clinical use is the availability of the risk factors required to compute the models [23]. Demographic and simple lifestyle risk factors may be available from clinical records, such as age, sex or smoking. However, models including genetic information or biomarkers require additional resources to collect and process samples. As the incidence of oral cancer is generally highest in middle- and lower-income countries [5], a strong case would need to be made for the use of these additional resources.

The models developed by Lee et al. [41] and Tota et al. [43], which use demographic and simple lifestyle variables, such as smoking status, and perform well in validations, may be relatively easy to implement within clinical settings. However, the collection of data on lifestyle risk factors in electronic health records is typically incomplete and varies significantly between countries and healthcare systems [54], including the documentation of tobacco use [55].

Models that use more detailed information about lifestyle behaviours, such as diet or family history, would require additional data collection (for example, a questionnaire within a routine consultation). This could be a barrier to implementation, given the costs and resources required. In the studies that have been identified in this review, there is no evidence that models containing more lifestyle variables (e.g., Chen 2022 [38], which includes diet) are more predictive of an oral cancer diagnosis than those with a small number of simpler variables of this type (e.g., Tota 2019a [43], which includes smoking and alcohol consumption).

Several clinical risk factors, including indicators of oral health, such as recurrent oral ulceration and regular dental visits, and HPV status (only for OPC), have been shown to be highly associated with oral cancer in previous studies [56]. We identified seven models [36,37,39,42,43] that include clinical risk factors, four of which would require the involvement of a clinician (and for HPV status, a blood test). For the reasons described above, comparison between these models is challenging. However, Tota et al. [43] show that the addition of HPV status improves model performance (from AUROC of 0.81 [2019a] to 0.87 [2019b]), indicating that testing for HPV may be a good use of limited resources when attempting to identify those at highest risk of oral cancer.

### 4.4. Recommendations

Currently, none of these models identified in this review can be recommended for use within a targeted screening programme. Although we identified several promising models, only one external validation (of two models developed by Tota et al. [43]) was found, and this study was assessed to be at high risk of bias. Future research should focus on determining the performance of these models in the populations where they are intended to be used (e.g., using population cohorts recruited in countries with high oral cancer incidence where screening programmes are being considered) and assessing the feasibility of implementation within the routine clinical practice.

We have shown that there is no evidence that many of the risk factors specific to oral cancer (e.g., indicators of oral health) improve model performance. Therefore, it may be reasonable to assess the identified models that used only demographic and lifestyle risk factors in cohorts that were not specifically recruited for oral cancer research. However, a key risk factor in literature [56] that is shown to improve model performance in this review [43] is HPV status, which may not be routinely available in nonspecific cohorts.

### 4.5. Strengths and Limitations

The systematic search of multiple electronic databases ensured good coverage of the existing literature for oral cancer risk prediction. However, our search was last updated in November 2022, so any articles published after this date are not included in this review. We further note the overlap with the recently published rapid review of prognostic models for head and neck cancer [24]; some of the models (*n* = 11) were identified in both reviews. However, we identified 12 additional models, including all six of the models containing genetic risk factors, that were not included in the rapid review (Appendix A).

We limited our review to English language papers; previous research has shown that this has minimal impact on the results of systematic review-based meta-analyses [57]. Nevertheless, the identified literature is more likely to overrepresent research carried out in English-speaking countries [58,59]. Ten non-English language articles were excluded (Figure 1, Appendix A) in this review. Nearly all were conducted in countries with a high incidence of oral cancer.

The use of the PROBAST assessment tool to assess the risk of bias enables the comparison of the identified studies across a range of different characteristics (including population and outcome), permitting not only an assessment of the quality of individual studies but also a way of identifying areas of weakness in the area as a whole.

The well-established inconsistency in the definition of oral cancer in literature [25] was also seen in this study and makes it challenging to establish direct comparisons between models that were developed for slightly different outcomes. For example, Fritsche et al. [46] included lip, oral cavity and pharynx as a single outcome for one of their models and only tongue cancer for the other. However, the aetiology, epidemiology and presentation of most oral cancers are similar [26,60], although there is a particularly strong association between OPC and HPV [61], and we did not identify any differences in performance between models, for example, developed for oral cavity and oropharyngeal cancers. Uncertainty about outcome definition (and other areas of heterogeneity) could be resolved by carrying out a head-to-head validation of all identified models in a single cohort for the same outcome.

## 5. Conclusions

This review has identified 23 risk prediction models for oral cancer with reported performance measures. Although two models by Tota et al. [43] were found to perform well (AUROC > 0.8) in external validation, all the identified studies had a high risk of bias and none of the models could be used in targeted screening programmes without further validation. With the heterogeneity of the studies, we, therefore, identify a need for high-quality external validations that compare the performance of existing models and public health modelling to assess the potential application to risk-stratified screening; we suggest prioritising these over further model development. There is currently no evidence that would justify the additional resources needed to include biomarkers or genetics in an assessment of oral cancer risk, although we note that there is limited research in these areas at present and that this may change in the future.

## Figures and Tables

**Figure 1 cancers-16-00617-f001:**
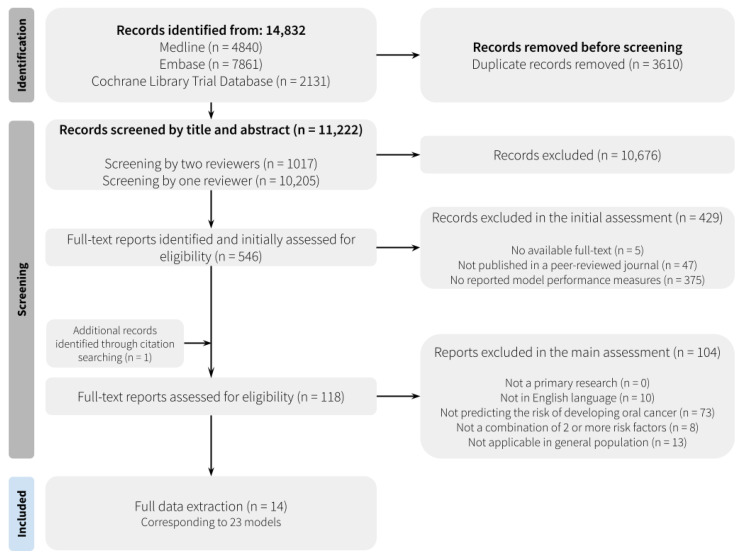
Preferred Reporting Items for Systematic Reviews and Meta-Analyses (PRISMA) flow diagram.

**Figure 2 cancers-16-00617-f002:**
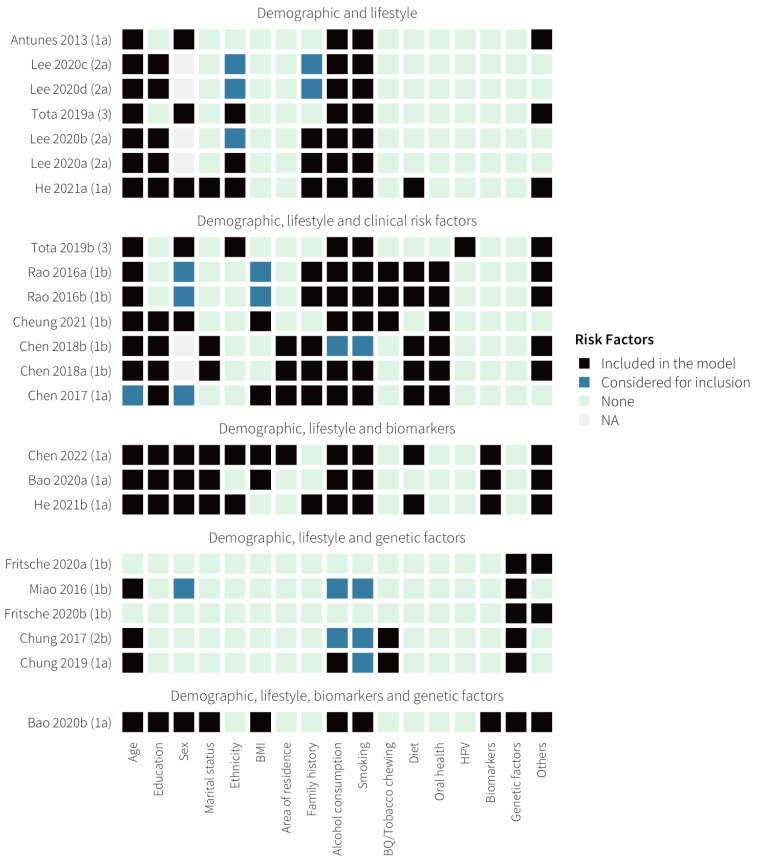
Considered and included risk factors in each model [34,35,36,37,38,39,40,41,42,43,44,45,46,47].

**Figure 3 cancers-16-00617-f003:**
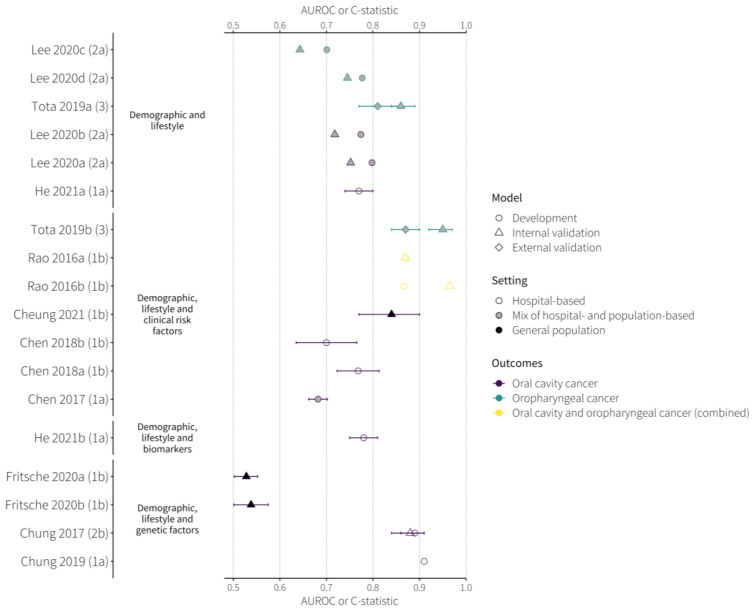
The reported area under the receiver operating characteristic curve (AUROC) values for the included models [34,35,36,37,38,39,40,41,42,43,44,45,46,47,48].

**Table 1 cancers-16-00617-t001:** Summary of risk prediction models (excluding models incorporating genetic variants).

First Author, Year	Country	Outcome ^a^	Age	Sex	Alcohol Consumption	Smoking	Clinical Risk Factors	Biomarkers	Other Lifestyle Risk Factors	Study Type	Study Setting	Tripod Level ^b^	Reported Performance Measures	Overall Risk of Bias	
														Development	Validation
Antunes, 2013 [34]	Brazil	OCC, OPC	X	X	X	X				CC	Hospital-based	1a	R^2^	High	Not applicable
Bao, 2020a [35]	China	OCC	X	X	X	X		X	X	CC	Hospital-based	1a	AIC	High	Not applicable
Chen, 2017 [36]	China	OCC			X	X	X		X	CC	Hospital-based cases and mixed controls	1a	AUROC	High	Not applicable
Chen, 2018a [37]	China	OCC (male)	X		X	X	X		X	CC	Hospital-based	1b	AUROC and calibration plot	High	High
Chen, 2018b [37]	China	OCC (female)	X				X		X	CC	Hospital-based	1b	AUROC and calibration plot	High	High
Chen, 2022 [38]	China	OCC	X	X	X	X		X	X	CC	Hospital-based	1a	Sens, Spec	High	Not applicable
Cheung, 2021 [39]	India	OCC	X	X	X	X	X		X	Cohort study based on a cluster RCT	General population	1b	AUROC, O/E ratio	High	High
He, 2021a [40]	China	OCC	X	X	X	X			X	CC	Hospital-based	1a	AUROC, AIC	High	Not applicable
He, 2021b [40]	China	OCC	X	X	X	X		X	X	CC	Hospital-based	1a	AUROC, AIC	High	Not applicable
Lee, 2020a [41]	United States	OCC (male)	X		X	X			X	CC	Hospital-based cases and mixed controls	2a	AUROC without CI	High	High
Lee, 2020b [41]	United States	OCC (female)	X		X	X				CC	Hospital-based cases and mixed controls	2a	AUROC without CI	High	High
Lee, 2020c [41]	United States	OPC (male)	X		X	X				CC	Hospital-based cases and mixed controls	2a	AUROC without CI	High	High
Lee, 2020d [41]	United States	OPC (female)	X		X	X				CC	Hospital-based cases and mixed controls	2a	AUROC without CI	High	High
Rao, 2016a [42]	India	OCC, OPC	X		X	X	X		X	CC	Hospital-based	1b	AUROC, Sens, Spec, PPV, NPV	High	High
Rao, 2016b [42]	India	OCC, OPC	X		X	X	X		X	CC	Hospital-based	1b	AUROC, Sens, Spec, PPV, NPV	High	High
Tota, 2019a [43]	United States	OPC	X	X	X	X			X	CC	Hospital-based cases and population-based controls	3	AUROC, O/E ratio in IV and EV	High	High
Tota, 2019b [43]	United States	OPC	X	X	X	X	X		X	CC	Hospital-based cases and population-based controls	3	AUROC, O/E ratio in IV and EV	High	High

Abbreviations: AIC, Akaike Information Criterion; AUROC, area under the receiver operating characteristic curve; CC, case-control; CI, confidence interval; EV, external validation; IV, internal validation; NPV, negative predictive value; OCC, oral cavity cancer; O/E, observed/expected; OPC, oropharyngeal cancer; PPV, positive predictive value; RCT, randomised controlled trial; Sens, sensitivity; Spec, specificity. ^a^ Each prediction model is for either a single or combined outcome. ^b^ Classification of prediction model according to the TRIPOD guidelines [30,31]: 1a, development only; 1b, development and validation using resampling; 2a, random split-sample development and validation; 3, development and validation using separate data.

**Table 2 cancers-16-00617-t002:** Summary of risk prediction models incorporating genetic variants.

First Author, Year	Country	Outcome ^a^	Genetic Factors	Non-Genetic Risk Factors	Study Type	Study-Setting	Tripod Level ᵇ	Reported Performance Measures	Overall Risk of Bias	
									Development	Validation
Bao, 2020b [35]	China	OCC	7 SNP ᶜ-constructed GRS	Selenium level	CC	Hospital-based	1a	AIC, OR	High	Not applicable
Chung, 2017 [44]	Taiwan	OCC	4 SNPs ᵈ	Age and betel quid chewing	CC	Hospital-based	2b	AUROC, Sens, Spec, OR	High	High
Chung, 2019 [45]	Taiwan	OCC	2 SNPs ᵉ	Age, betel quid chewing and alcohol consumption	CC	Hospital-based	1a	AUROC	High	High
Fritsche, 2020a [46]	United Kingdom (UK Biobank)	OCC	1,119,238 SNPs	EHR-derived phenotypes	CC	General population	1b	AUROC, R^2^, Brier score, OR	High	High
Fritsche, 2020b [46]	Finland (FinnGen)	OCC ᶠ	931,954 SNPs	EHR-derived phenotypes	CC	General population	1b	AUROC, R^2^, Brier score, OR	High	High
Miao, 2016 [47]	China	OCC	3 SNPs	Age	CC	Hospital-based	1a	Balance accuracy	High	Not applicable

Abbreviations: AIC, Akaike Information Criterion; AUROC, area under the receiver operating characteristic curve; CC, case-control; EHR, electronic health record; GRS, genetic risk score; OCC, oral cavity cancer; OR, odds ratio; Sens, sensitivity; SNP, single nucleotide polymorphism; Spec, specificity. ^a^ Each prediction model is for either a single- or combined-outcome. ^b^ Classification of prediction model according to the TRIPOD guidelines [30,31]: 1a, development only; 1b, development and validation using resampling; 2b, nonrandom split-sample development and validation; ^c^ cIncluded SNPs: rs1800668, rs3746165, rs7310505, rs4964287, rs9605030, rs3788317, rs13054371. ^d^ Included SNPs: rs2070833, rs550675, rs139994842, rs2822641. ^e^ Included SNPs: rs550675, rs28647489. ^f^ Tongue cancer.

**Table 3 cancers-16-00617-t003:** List of the risk factors in the included models.

Risk Factor, Category	Considered	Included	Risk Factor, Category	Considered	Included
Demographic and lifestyle			Beans and/or soy products	3	2
Personal characteristics			Tea consumption	2	2
Age	20	21	Spicy foods	2	2
Education level	14	14	Poultry/domestic meat	3	1
Sex	13	9	Milk and dairy products	3	1
Marital status	7	7	Pickled food	3	1
Ethnicity	9	6	Processed meat	1	1
BMI	7	5	Tea concentration	2	0
Area of residence	5	5	Tea types	2	0
Occupation *	3	3	Tea temperature	2	0
Lifetime number of sexual partners *	2	2	Tobacco/BQ chewing	5	5
Age of first intercourse *	2	1	Status of tobacco/BQ chewing	4	4
Parental education level *	2	0	Duration of tobacco/BQ chewing	1	1
Socioeconomic condition *	2	0	Intensity of tobacco/BQ chewing	1	1
Alcohol consumption	21	18	Past user of tobacco/BQ chewing	1	1
Alcohol consumption status	12	10	Clinical risk factors		
Intensity of alcohol consumption	9	9	Oral health or oral habits	6	6
Parental alcohol consumption status	2	0	Teeth loss	2	2
Duration of alcohol consumption	1	1	Recurrent oral ulceration	3	3
Smoking	21	17	Regular dental visit	2	2
Smoking status/history	15	12	Denture wearing	3	1
Intensity of tobacco/cigarette smoking	9	8	Frequency of tooth-brushing	2	0
Duration of tobacco/cigarette smoking	5	5	Mouth rinsing habit	2	2
Passive smoking	2	1	Oral cancer screening status	1	1
Family history	9	7	HPV status	1	1
Family history of head and neck cancer	6	4	Genetic factors	6	6
Family history of any cancer	5	5	Genetic risk score	3	3
Diet	8	8	Biomarkers	3	3
Vegetables (leafy and/or other)	6	6	Arsenic level	1	1
Fish	6	5	Selenium level	1	1
Seafood	6	5	Cerium level	1	1
Fruits	6	5	Other risk factors	4	3
Eggs	5	3	EHR-derived phenotype	2	2
Red meat	5	3	Cooking oil fume exposure	2	1

Abbreviations: BMI, body mass index; BQ, betel quid; HPV, human papillomavirus. * Five personal characteristics risk factors with the least occurrence were categorised as ‘Others’ in Figure 3.

**Table 4 cancers-16-00617-t004:** Summary details of performance measure.

First Author, Year	Development				Validation ^a^			
	Discrimination	Calibration	Accuracy	Other Measures	Discrimination	Calibration	Accuracy	Other Measures
**Models without genetic variants**								
Antunes, 2013 [34]				Pseudo-R^2^ = 0.186				
Bao, 2020a [35]		AIC: 542.846						
Chen, 2017 [36]	AUROC: 0.682 (95% CI: 0.662–0.702)							
Chen, 2018a [37]	C-index: 0.768 (95% CI: 0.723–0.813)	Calibration plot: shows good calibration						
Chen, 2018b [37]	C-index: 0.700 (95% CI: 0.635–0.765)	Calibration plot: shows good calibration						
Chen, 2022 [38]			Sens: 69.9% Spec: 91.4%					
Cheung, 2021 [39]					C-index: 0.84 (95% CI: 0.77–0.90)	O/E ratio: 1.08 (95% CI: 0.81–1.44)		
He, 2021a [40]	AUROC: 0.77 (95% CI: 0.74–0.80)	AIC: 1040.50						
He, 2021b [40]	AUROC: 0.78 (95% CI: 0.75–0.81)	AIC: 1033.82						
Lee, 2020a [41]	AUROC: 0.798				AUROC: 0.752	Calibration plot by decile: shows good calibration		
Lee, 2020b [41]	AUROC: 0.774				AUROC: 0.718	Calibration plot by decile: shows good calibration		
Lee, 2020c [41]	AUROC: 0.701				AUROC: 0.643	Calibration plot by decile: shows overestimation in the three highest deciles		
Lee, 2020d [41]	AUROC: 0.777				AUROC: 0.745	Calibration plot by decile: shows good calibration		
Rao, 2016a [42]	AUROC: 0.870		Sens: 74.6% Spec: 84.6% PPV: 76.7% NPV: 83.0%		AUROC: 0.869		Sens: 74.4% Spec: 85.1% PPV: 77.3% NPV: 83.0%	
Rao, 2016b [42]	AUROC: 0.866		Sens: 92.8% Spec: 60.3% PPV: 60.7% NPV: 92.7%		AUROC: 0.865		Sens: 96.6% Spec: 39.3% PPV: 51.4% NPV: 94.6%	
Tota, 2019a [43]					Internal validation: AUROC: 0.86 (95% CI: 0.84–0.89) External validation: AUROC: 0.81 (95% CI: 0.77–0.86)	Internal validation: Overall O/E ratio: 1.01 (95% CI: 0.70–1.32) External validation: O/E ratio: 1.08 (95% CI: 0.77–1.39)		
Tota, 2019b [43]					Internal validation: AUROC: 0.95 (95% CI: 0.92–0.97) External validation: AUROC: 0.87 (95% CI: 0.84–0.90)	Internal validation: O/E ratio: 1.01 (95% CI: 0.70–1.32) External validation: O/E ratio: 1.08 (95% CI: 0.77–1.39)		
**Models incorporating genetic variants**								
Bao, 2020b [35]		AIC: 504.162		Adjusted OR for GRS: 0: Reference 1: 1.908 (95% CI: 1.086–3.352) 2: 1.940 (95% CI: 1.055–3.567)				
Chung, 2017 [44]	AUROC: 0.89 (95% CI: 0.86–0.91)		Sens: 86.7% Spec: 86%	Adjusted OR for GRS: 0: Reference 1: 0.96 (95% CI: 0.60–1.54) 2: 1.29 (95% CI: 0.79–2.10) 3: 1.31 (95% CI: 0.60–2.85) 4: 3.11 (95% CI: 1.21–10.67)	AUROC: 0.88 (95% CI: 0.84–0.91)		Sens: 86.3% Spec: 86.5%	
Chung, 2019 [44]	AUROC: 0.91		Sens: 85.7% Spec: 85.7%	Adjusted OR for GRS: 0: Reference 1: 1.68 (95% CI: 1.01–2.81) 2: 6.12 (95% CI: 1.66–22.49)				
Fritsche, 2020a [46]				OR Top 1% vs. other: 1.63 (95% CI: 0.812–3.26) R^2^: 0.00207	AUROC: 0.528 (95% CI: 0.502–0.552)	Brier score: 0.0829		
Fritsche, 2020b [46]				OR Top 1% vs. other: 1.69 (95% CI: 0.61–4.71) R^2^: 0.00325	AUROC: 0.538 (95% CI: 0.501–0.575)	Brier score: 0.0827		
Miao, 2016 [47]			Training balance accuracy: 0.8221 Testing balance accuracy: 0.5491					

Abbreviations: AIC, Akaike Information Criterion; AUROC, area under the receiver operating characteristic curve; O/E, observed/expected; OR, odds ratio; Sens, sensitivity; Spec, specificity. ᵃ All reported validation measures are internal validation except for Tota et al. [43], which reported internal and external validation.

## Data Availability

Data about the systematic review is available upon requests from the corresponding author. Details of the individual models are available in the original studies.

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
