# Peer review of "Risk Prediction Models for Oral Cancer: A Systematic Review"

_cancers, 2024, doi:10.3390/cancers16030617_

Round 1

Reviewer 1 Report

Comments and Suggestions for Authors

Dear editor and author,

This is a well-prepared review that aims to systematically map the published models that predict the development of oral cancer and are suitable for use in the general population, describing and comparing the identified models, focusing on their development, including risk factors, performance, and applicability to risk-stratified screening. It presents a clear method and adequate processing of the data for a definite outcome. However, it is noted that the manuscript still needs modifications concerning the flaws mentioned below:

 1.      Please update the information in PROSPERO (CRD42022316516), which wrote “The literature search will be performed in MEDLINE, Embase and the Cochrane Library without publication year limit (beginning of database to January 2022). Language restriction will not be applied to the search strategy, however, it will be used in the eligibility criteria for study selection.” The date is not the same with what is mentioned in the manuscript, as well as other details published on the website (https://www.crd.york.ac.uk/PROSPERO/display_record.php?RecordID=316516).

2.      It has already the end of the year 2023, but the database ends in Nov 2022, I believe there are many publications during this year. The author shall update the database in t in the Medline, Embase and Cochrane Library.

3.      As there are more than ten thousand publications this review included, how the authors manage so many publications, do they apply literature management software such as Endnote, and how two reviewers cooperate to finalize the suitable articles.

4.      “Participants/population

Inclusion: Individuals older than 18 years of age, in the general population, with no history of oral cancer.Exclusion: Individuals with a history of oral cancer.”

Why the individuals under 18 years excluded? Could the author make a subgroup to discuss the risk model on minors with oral cancer

5.      Also, the author did not mention that oral cancer is a primary tumor or metastatic tumor which can be a neglectable factor in the results.

6.      In Table One, there are only 14 articles were included, which are from America and Asia, however, no research is from European countries, which is astonishing as there are so many reputational institutes and hospitals working on oral cancer, could the author check the final result again to verify the process?

7.      Could the author explain why the PROBAST assessment tool is applied to the bias assessment?

8.      The models were wide-ranging (AUROCs 0.53-0.95), how to define the AUROCs value? How to improve the AUROCS, and how to explain the wide range of AUROCs?

9.      The Conclusion is lengthy which makes it unclear for the readers to grasp the key of this review, could the author simplify this part?

10.   How does this finding guide the oral and maxillofacial practitioners or clinicians to diagnose, treat, and evaluate this disease?  

Reviewer 2 Report

Comments and Suggestions for Authors

1. The authors' definition for "oral cancer" is wrong. Oral cancer does not include "oropharyngeal cancer  (OPC)". If the authors wish to include OPC the title should include OPC.

2. The term "prevalence" is incorrectly used, This is a common mistake made. The correct term is "incidence".

3. The objective of this review is to recommend few "risk factor models" for future use out of the 23 models identified in the search. For this the authors should do a multivariate analysis and identify a few that are operationally better [i.e best buys]  to identify high risk subjects.

4. The authors could also provide a Forrest plot to illustrate significantly better models.

5. Models for oral cancer screening should be separately discussed to OPC screening. OC and OPC have very different risk factors [OPC are mainly caused by HPV and only a minor proportion (about 5%)  of OC s are caused by HPV] and the two groups also very different demographic factors.

6. Fritsche et al studies are listed by the authors as United Kingdom / General population. But these studies were done from biobanks. In Table 2 this should be clarified.

7. The conclusions should state the best models for future use/ experiments. 

Comments on the Quality of English Language

none

Round 2

Reviewer 1 Report

Comments and Suggestions for Authors

Dear Authors,

I did not the response letter to review one, please check and answer review ones' questions.

Comments on the Quality of English Language

Dear Authors,

I did not the response letter to review one, please check and answer review ones' questions.

Reviewer 2 Report

Comments and Suggestions for Authors

None

Comments on the Quality of English Language

none

Round 3

Reviewer 1 Report

Comments and Suggestions for Authors

Dear Authors

Thank you for your modification, most of the answers are satisfactory, however, the answer to question 2  ("We further note that our search ends in November 2022; given the fast-moving nature of this field of research, we expect that a small number of eligible studies have been published since this date."

) is overly subjective.

If they believe only a few studies published within one year, they need to provide enough evidence to support their conjecture. Otherwise, they need to check them in PubMed and other databases.

Round 4

Reviewer 1 Report

Comments and Suggestions for Authors

Thank you for your modification.

The author need to check the articles published before 2024 Jan in PubMed and other databases, otherwise, the review is outdate for Cancers Journal.
